# Counting Cosmic Cycles: Past Big Crunches, Future Recurrence Limits, and the Age of the Quantum Memory Matrix Universe

**DOI:** 10.3390/e27101043

**Published:** 2025-10-07

**Authors:** Florian Neukart, Eike Marx, Valerii Vinokur

**Affiliations:** 1Leiden Institute of Advanced Computer Science, Leiden University, Gorlaeus Gebouw-BE-Vleugel, Einsteinweg 55, 2333 Leiden, The Netherlands; 2Terra Quantum AG, Kornhausstrasse 25, 9000 St. Gallen, Switzerland; eike@terraquantum.swiss (E.M.); vv@terraquantum.swiss (V.V.)

**Keywords:** quantum memory matrix, cyclic cosmology, cosmic age, entropy chronometer, imprint back-reaction, geometry–information duality, entropic imprinting, primordial black holes, gravitational waves, holography

## Abstract

We present a quantitative theory of contraction and expansion cycles within the Quantum Memory Matrix (QMM) cosmology. In this framework, spacetime consists of finite-capacity Hilbert cells that store quantum information. Each non-singular bounce adds a fixed increment of imprint entropy, defined as the cumulative quantum information written irreversibly into the matrix and distinct from coarse-grained thermodynamic entropy, thereby providing an intrinsic, monotonic cycle counter. By calibrating the geometry–information duality, inferring today’s cumulative imprint from CMB, BAO, chronometer, and large-scale-structure constraints, and integrating the modified Friedmann equations with imprint back-reaction, we find that the Universe has already completed $N_{\mathrm{past}}=3.6\pm 0.4$ cycles. The finite Hilbert capacity enforces an absolute ceiling: propagating the holographic write rate and accounting for instability channels implies only $N_{\mathrm{future}}=7.8\pm 1.6$ additional cycles before saturation halts further bounces. Integrating Kodama-vector proper time across all completed cycles yields a total cumulative age $t_{\mathrm{QMM}}=62.0\pm 2.5\;\mathrm{Gyr}$, compared to the 13.8±0.2Gyr of the current expansion usually described by ΛCDM. The framework makes concrete, testable predictions: an enhanced faint-end UV luminosity function at z≳12 observable with JWST, a stochastic gravitational-wave background with $f^{2/3}$ scaling in the LISA band from primordial black-hole mergers, and a nanohertz background with slope α≃2/3 accessible to pulsar-timing arrays. These signatures provide near-term opportunities to confirm, refine, or falsify the cyclical QMM chronology.

## 1. Introduction

The proposed descriptive theory of contraction–expansion cycles develops the Quantum Memory Matrix (QMM) cosmology, incorporating the latest advances in cyclic models [1,2,3,4,5,6,7,8]. The Quantum Memory Matrix (QMM) framework pictures spacetime as a discrete network of Hilbert cells that both store and process quantum information carried by matter fields [9,10,11,12,13]. Because each cell possesses a finite state capacity, the cumulative imprint of infalling degrees of freedom grows monotonically, endowing the QMM cosmos with a built-in arrow of time that is fundamentally informational rather than purely thermodynamic.

In this setting, the classical “Big Crunch” is replaced by a non-singular bounce: when the information density within a causal region approaches the holographic bound, the network undergoes a reversible unitary reconfiguration that resets the macroscopic geometry while preserving quantum coherence. Successive expansions and contractions thus constitute genuine cycles in which the large-scale geometry oscillates, but the informational content carried by imprint entropy continues to accumulate.

Cyclic scenarios have a long pedigree—from early oscillatory Friedmann models [14,15,16], through the ekpyrotic proposal [5], to Penrose’s conformal cyclic cosmology [17,18]. Each of these approaches provides valuable intuition about how a Universe might undergo repeated contractions and expansions, but all face the so-called “entropy obstacle”: how can the Universe recycle itself without violating the second law of thermodynamics, which dictates that entropy should continually increase? Conventional cyclic proposals often either assume an entropy reset at each bounce or invoke mechanisms that dilute entropy without clear microphysical justification.

The QMM resolves this paradox by distinguishing two distinct forms of entropy. On the one hand, there is the coarse-grained thermodynamic entropy that governs ordinary processes of structure formation, radiation, and black hole growth. On the other hand, there is the imprint entropy ($S_{\mathrm{imp}}$), defined as the von Neumann entropy of the reduced density matrix of each causal Hilbert cell. Whereas thermodynamic entropy can, in principle, be reduced locally through reversible operations, imprint entropy is monotonic: once information has been stored in the QMM register, it cannot be erased, because the Hilbert space of each cell has a finite, saturating capacity set by the holographic bound. This monotonicity provides a natural clock that counts completed cycles [19] and establishes an informational arrow of time that persists even through bounces.

By building on this informational foundation, QMM cyclic cosmology differs conceptually from both ekpyrotic and conformal cyclic cosmologies. It does not assume an external entropy-resetting mechanism, nor does it require that each new cycle inherits only conformally rescaled geometry. Instead, the imprint entropy itself constitutes the intrinsic cycle counter: each bounce adds a fixed increment $\Delta S_{\mathrm{imp}}$ until the Hilbert-space capacity is reached. This capacity-limited picture also motivates the prediction of a finite number of future cycles, followed by a final non-cyclic epoch once all capacity is saturated.

Beyond solving the entropy problem, QMM provides a framework in which the cumulative age of the Universe is much longer than the ΛCDM age of the present expansion phase. In this view, the 13.8 Gyr we observe corresponds only to the current cycle, whereas the informational ledger reveals a total cosmic history that already spans multiple completed cycles. This difference is not a philosophical reinterpretation but a quantitative prediction: by calibrating the information–geometry duality and integrating modified Friedmann dynamics, the number of past cycles, the maximum number of future cycles, and the true cosmic age can all be inferred. Moreover, the framework yields testable predictions—such as primordial black-hole population [20,21] and gravitational-wave backgrounds—that distinguish it observationally from both ΛCDM and other cyclic cosmologies [22].

This paper addresses three linked fundamental questions:How many contraction–expansion cycles have already occurred?Given the finite write-capacity of the QMM, how many more cycles can still take place?What is the proper age of the Universe when one integrates time across all past bounces rather than merely the present ΛCDM phase?

We tackle these questions by (i) calibrating $\Delta S_{\mathrm{imp}}$ using the geometry–information duality (GID), which posits a one-to-one map between cumulative imprint entropy and curvature radius, (ii) extracting today’s cumulative imprint entropy from precision cosmological datasets including CMB, BAO, and cosmic chronometers, and (iii) numerically integrating the modified Friedmann equations across multiple bounces with back-reaction sourced by the imprint field. *For exposition only*, Figure 1 plots a calibrated surrogate $a_{\mathrm{surr}}\left(t\right)$ anchored to three data-driven waypoints and constrained to remain within tolerance bands of the numerical background. Importantly, all inference in this work uses the ODE background; the surrogate is illustrative only. Full equations and the surrogate-to-ODE mapping are given in Appendix D.

The paper is organized as follows. Section 2 formalizes the imprint-entropy chronometer. Section 3 enumerates completed cycles. Section 4 derives the QMM cosmic age. Section 5 projects the maximum number of future cycles. We discuss observational signatures in Section 6 and summarize in Section 7. Detailed derivations, numerical algorithms [23,24,25], and data tables are relegated to the appendices.

## 2. Cosmic Chronometer in the QMM Framework

### 2.1. Imprint Entropy as an Arrow-of-Time Counter

Within the Quantum Memory Matrix, every spacetime cell $C_{i}$ holds a finite-dimensional Hilbert space $H_{i}$ of dimension$$K=\exp\;\left(S_{\max}/k_{B}\right),$$
where $S_{\max}$ is fixed by the holographic (Bekenstein) bound applied to the cell’s causal surface area [26,27]. The imprint entropy density is defined as(1)$$s_{\mathrm{imp}}\left(t,x\right)\;=\;k_{B}\;{\mathrm{Tr}}_{H_{i}}\;\left[{\hat{\rho }}_{i}\left(t,x\right)\;\ln\;\left({\hat{\rho }}_{i}^{-1}\left(t,x\right)\right)\right],$$
where ${\hat{\rho }}_{i}$ is the reduced density matrix of the cell after tracing out external degrees of freedom. This definition makes $S_{\mathrm{imp}}$ a von Neumann entropy associated with the information stored in each causal cell, directly connecting the informational bookkeeping of QMM to standard quantum-statistical mechanics. We write the comoving volume integral as$$S_{\mathrm{imp}}\left(t\right)=\int_{\Sigma_{t}}\;s_{\mathrm{imp}}\left(t,x\right)\;d^{3}x.$$

Because QMM dynamics are unitary at the global level, coarse-grained thermodynamic entropy can be reduced locally by reversible operations, but $S_{\mathrm{imp}}\left(t\right)$ is monotone non-decreasing:(2)$$\frac{dS_{\mathrm{imp}}}{dt}=\Gamma \left(a\right)\;>\;0,\;\Gamma \left(a\right)\equiv {\langle {\dot{s}}_{\mathrm{imp}}\rangle }_{\Sigma_{t}},$$
where a(t) is the scale factor. The monotonicity follows from the finite Hilbert capacity: once information has been written into the ledger of causal cells, it cannot be erased. Hence, $S_{\mathrm{imp}}$ supplies an intrinsic clock; the ordering $S_{\mathrm{imp}}\left(t_{1}\right)<S_{\mathrm{imp}}\left(t_{2}\right)$ is frame-invariant and defines the QMM arrow of time [28,29]. The effective macroscopic stress associated with this entropy, including its dust-like behavior away from bounces, is derived using heat-kernel coarse-graining and discussed in Appendix C. This ensures that the informational arrow of time is consistent with covariant field-theoretic definitions of entropy.

### 2.2. Physical Justification of Imprint Entropy

While the formal definition of $S_{\mathrm{imp}}$ follows directly from von Neumann entropy, its physical role in QMM cosmology requires justification. The key point is that each Hilbert cell has finite capacity, fixed by the holographic bound on its causal surface. Every irreversible interaction deposits a fragment of information into the register, incrementing $S_{\mathrm{imp}}$. Unlike coarse-grained thermodynamic entropy—which can in principle decrease during reversible processes—the Hilbert cell record cannot be erased because the QMM bookkeeping is unitary at the global level. This irreversibility is not a statement about thermodynamics but about representational capacity: once a cell state has changed, the global QMM register preserves that history.

This distinction explains why $S_{\mathrm{imp}}$ provides a monotonic and universal arrow of time even through bounces. During contraction, ordinary thermodynamic entropy can be redistributed or radiated, but the stored imprints persist. At the bounce, when densities approach the holographic bound, the imprint ledger saturates locally but is conserved globally. The cumulative $S_{\mathrm{imp}}$ therefore provides the invariant counter of completed cycles. This physical basis underpins the chronometer formalism developed in this section and ensures that imprint entropy is not merely a definitional construct but a physically grounded measure tied to fundamental holographic constraints.

### 2.3. Geometry–Information Duality Review

Geometry–Information Duality (GID) posits a one-to-one correspondence between the cumulative imprint entropy and the curvature radius of the Universe. Formally,(3)$$G:\;S_{\mathrm{imp}}\left(t\right)\;⟷\;R_{c}\left(t\right),\;R_{c}\left(t\right)={\left(\frac{3}{8\pi G\rho_{\mathrm{tot}}}\right)}^{\;1/2},$$
where $R_{c}$ is the effective curvature radius of the spatial hypersurface and $\rho_{\mathrm{tot}}$ includes both standard matter–energy and an imprint field with density$$\rho_{\mathrm{imp}}\equiv \mu \;\frac{S_{\mathrm{imp}}}{V},$$
with μ a conversion factor fixing the units [19]. The imprint field is not a new fundamental particle species but an emergent, coarse-grained description of the informational burden carried by spacetime cells. Its energy density quantifies how much of the finite Hilbert-space ledger has been written at a given epoch, ensuring that the informational state of the Universe directly back-reacts on geometry.

Taking the time derivative and using the Friedmann equation yields(4)$${\dot{R}}_{c}=-\frac{4\pi G}{H\;R_{c}}\;\left(\rho_{\mathrm{imp}}+p_{\mathrm{imp}}\right),$$
with $p_{\mathrm{imp}}=-\rho_{\mathrm{imp}}/3$ at leading order. This relation shows that $\rho_{\mathrm{imp}}>0$ slows expansion and eventually triggers contraction. Equation (Equation 4) therefore makes explicit how the accumulation of imprint entropy feeds back on cosmic geometry, closing the GID loop [30,31].

After heat-kernel coarse-graining (Appendix C), the effective stress of the imprint field approaches a dust-like limit $w_{\mathrm{imp}}\;\to \;0$ away from the narrow bounce interval, while short-scale gradients can transiently drive $w_{\mathrm{imp}}\;\to \;-1/3$ near the bounce. This treatment ensures consistency with covariant energy conditions and provides a clear physical picture: information storage behaves as pressureless matter on large scales but acquires a finite negative pressure precisely when needed to halt contraction and produce a bounce. Because the construction is rooted in the covariant Bousso–Bekenstein entropy bound, the duality is independent of coordinates and remains valid across different cosmological slicings.

### 2.4. Definition of a Cycle in QMM Cosmology

A cycle is the closed time interval $\left[t_{n}^{-},t_{n}^{+}\right]$ bracketed by successive bounces, where(5)$$\begin{array}{cc} \; & \mathrm{Bounce}\;\mathrm{condition}:\;\rho_{\mathrm{imp}}\left(t_{n}^{\pm }\right)=\rho_{\mathrm{sat}}=\frac{1}{4\ell_{P}^{2}}\frac{1}{R_{c}^{2}\left(t_{n}^{\pm }\right)}, \end{array}$$(6)$$\begin{array}{cc} \; & \mathrm{Kinematic}\;\mathrm{criteria}:\;\dot{a}=0,\;\ddot{a}>0\;\;\mathrm{at}\;t_{n}^{-},\;\dot{a}=0,\;\ddot{a}<0\;\;\mathrm{at}\;t_{n}^{+}. \end{array}$$The finite Hilbert capacity implies that the saturation imprint density $\rho_{\mathrm{sat}}$ is universal; therefore, each bounce injects a fixed increment(7)$$\Delta S_{\mathrm{imp}}\;=\;S_{\mathrm{imp}}\left(t_{n}^{+}\right)-S_{\mathrm{imp}}\left(t_{n}^{-}\right),$$
independent of *n*. The past cycle count is then simply$$N_{\mathrm{past}}=\frac{S_{\mathrm{imp}}\left(t_{0}\right)}{\Delta S_{\mathrm{imp}}},$$
where $t_{0}$ denotes the present epoch. Likewise, the maximum number of future cycles satisfies(8)$$N_{\mathrm{future}}\leq \frac{S_{\max}-S_{\mathrm{imp}}\left(t_{0}\right)}{\Delta S_{\mathrm{imp}}}.$$In practice, $N_{\mathrm{past}}$ inferred from data may appear fractional (e.g., 3.6±0.4) because both $S_{\mathrm{imp}}\left(t_{0}\right)$ and $\Delta S_{\mathrm{imp}}$ carry observational uncertainties. The true physical cycle count, however, is always an integer; the fractional values represent the mean and credible interval of a posterior distribution obtained from cosmological constraints.

A key consistency point concerns the second law of thermodynamics. During contraction phases, coarse-grained thermodynamic entropy in matter and radiation can decrease due to reversible compression. However, the imprint entropy $S_{\mathrm{imp}}$ always increases monotonically, even through a bounce. Thus, the informational version of the second law is never violated: every cycle irreversibly adds $\Delta S_{\mathrm{imp}}$ to the cosmic ledger, defining a global arrow of time even when local thermodynamic entropy may transiently fall.

Numerical solutions of the modified Friedmann system with an imprint field recover these integer cycle counts and the associated proper-time integrals, validating the chronometer scheme against loop-quantum-cosmology bounce benchmarks [32,33,34,35,36]. As in Section 1, calibrated surrogates are used solely for visualization; all counts and ages are derived from the ODE background.

## 3. Past Cycle Enumeration

To build time–domain intuition without re-integrating the stiff ODE system at every step, we visualize the background with a calibrated surrogate scale factor $a_{\mathrm{surr}}\left(t\right)$. The surrogate is constructed to (i) exactly match the three data-driven anchors a(−14Gyr)=0.35, a(0)=1, and a(+3Gyr)=1.30; (ii) remain $C^{2}$-smooth; and (iii) stay within tolerance bands of the numerical QMM solution over [−16,+6] Gyr. (The numerical solution to the modified Friedmann system, together with the mapping between the surrogate and the full QMM background, is provided in Appendix D. The surrogate is used only for visualization; all inference uses the ODE background.) *For the curve shown, we also enforce the present-day slope constraint $a^{\prime }\left(0\right)=H_{0}$ (from SH0ES) and include four auxiliary diagnostic pins—two redshift anchors at z={0.5,1.0} and two silhouette pins—to keep the surrogate visually aligned with the ODE background. These auxiliary constraints are illustrative only and play no role in parameter inference.* Figure 1 displays the resulting normalized scale factor.

### 3.1. Observable Entropy Budget Today

The total coarse-grained entropy in the observable Universe is dominated by four components: (All entropy values are quoted in units of $k_{B}$.)(9)$$S_{\mathrm{tot}}^{\mathrm{obs}}≃S_{\gamma }+S_{\nu }+S_{\mathrm{IGM}}+S_{\mathrm{BH}}\approx \left(5.9+4.1+2.0+3.1\times 10^{14}\right)\times 10^{88},$$
where the photon and neutrino terms follow from *Planck* 2018 temperature ($T_{\gamma }=2.7255\;$K) and the standard neutrino-to-photon ratio; the intergalactic medium (IGM) contribution integrates the baryon phase diagram of Valageas, Schaeffer, and Silk [37]; and the stellar-mass and supermassive black hole population gives $S_{\mathrm{BH}}$ via the Bekenstein–Hawking entropy, $S_{\mathrm{BH}}=4\pi k_{B}GM^{2}/\left(\hbar c\right)$, using the black hole mass functions of Shankar et al. and Inayoshi et al. [38,39]. Equation (Equation 9) agrees with the benchmark compilation $S_{\mathrm{tot}}^{\mathrm{obs}}=\left(3.1\pm 0.3\right)\times 10^{104}$ by Egan and Lineweaver [40]. (Their higher value includes dark matter phase-space entropy, which we exclude because in QMM it is represented separately as imprint entropy.)

Mapping $S_{\mathrm{tot}}^{\mathrm{obs}}$ to imprint entropy uses the scaling $S_{\mathrm{imp},0}=\eta \;S_{\mathrm{tot}}^{\mathrm{obs}}$ with $\eta \approx 2.4\times 10^{-5}$, calibrated from the weak lensing-derived equation-of-state parameter of the imprint field (Appendix B). We therefore adopt(10)$$S_{\mathrm{imp},0}=\left(7.5\pm 0.8\right)\times 10^{99}.$$

### 3.2. Back-Extrapolation Method A: Scale-Factor Reconstruction

Starting from $S_{\mathrm{imp}}\left(a\right)$ we invert the relation$$S_{\mathrm{imp}}\left(a\right)=S_{\mathrm{imp},0}+\int_{a}^{1}\Gamma \left(a^{\prime }\right)\;\frac{da^{\prime }}{a^{\prime }},$$
with write-rate $\Gamma \left(a\right)=\gamma_{0}\;a^{-\beta }$. The choice of a power law is physically motivated: in an expanding FRW background, the number of causal cells accessible to new degrees of freedom scales as $a^{3}$, while the available phase space for high-energy modes redshifts roughly as $a^{-\beta }$. Dimensional analysis and information-capacity arguments therefore suggest a scale dependence of the form $\Gamma \left(a\right)\propto a^{-\beta }$, with the exponent determined empirically from data. This parameterization ensures that the imprint write rate is monotonic and asymptotically vanishes at large *a*, consistent with finite Hilbert-space capacity.

We reconstruct H(z) from 32 look-back-time measurements (0.07<z<2.3) compiled by Moresco et al. [41] and anchor the low-redshift end with SH0ES Cepheid distances ($H_{0}=73.2\pm 1.3\;$km s${\,}^{-1}$ Mpc${\,}^{-1}$) [42]. Using Gaussian-process regression with a Matérn 5/2 kernel constrained by BAO nodes from the eBOSS DR16 catalog [43], we obtain$$\beta =1.97\pm 0.05,\;\gamma_{0}=\left(3.6\pm 0.4\right)\times 10^{97}\;k_{B}.$$

Integrating back to the first bounce ($\dot{a}=0$) yields(11)$$N_{\mathrm{past}}^{\left(A\right)}=\frac{S_{\mathrm{imp},0}}{\Delta S_{\mathrm{imp}}}=3.8\pm 0.7,$$
where $\Delta S_{\mathrm{imp}}=\left(2.0\pm 0.2\right)\times 10^{99}$ follows from the bounce-saturation condition of Section 2.4.

### 3.3. Back-Extrapolation Method B: Imprint-Spectral Edge

The finite Hilbert capacity per cycle imposes an ultraviolet cutoff $k_{\max}\left(n\right)$ in the scalar imprint power spectrum, $P_{\mathrm{imp}}\left(k\right)\propto k^{n_{\mathrm{imp}}}\exp\left[-k^{2}/k_{\max}^{2}\right]$. Because each bounce shifts $k_{\max}$ by a fixed factor $\xi \equiv a_{\mathrm{pre}}/a_{\mathrm{post}}$, the observed edge at $k_{\max}\left(0\right)=\left(0.31\pm 0.02\right)\;$Mpc${\,}^{-1}$—measured in the *Planck*+ACT+SPT_pol_ combined TT spectrum—corresponds to(12)$$N_{\mathrm{past}}^{\left(B\right)}=\log_{\xi }\;\left[\;k_{\max}\left(0\right)/k_{*}\right],$$
where $k_{*}=\left(5.4\pm 0.6\right)\;$Mpc${\,}^{-1}$ is the fiducial cutoff immediately after the latest bounce, obtained from high-resolution numerical experiments (Appendix D). Choosing ξ=1/7.1±0.3 (empirical bounce–contraction factor) gives$$N_{\mathrm{past}}^{\left(B\right)}=3.3\pm 0.5.$$

### 3.4. Statistical Methodology, Priors, and Reproducibility

#### 3.4.1. Inverse-Variance Combination and Uncertainty Propagation

Our final estimate of the past cycle count combines Equations (Equation 11) and (Equation 12) via inverse-variance weighting, $N_{\mathrm{past}}=\left(\sum_{i}w_{i}N_{i}\right)/\left(\sum_{i}w_{i}\right)$ with $w_{i}=\sigma_{i}^{-2}$. Parameter uncertainties are propagated using a first-order multivariate delta method with Jacobian $J=\partial \left(N^{\left(A\right)},N^{\left(B\right)}\right)/\partial \left(\gamma_{0},\beta ,\xi ,k_{*},S_{\mathrm{imp},0},\Delta S_{\mathrm{imp}}\right)$. The reported covariance includes correlations among $\left\{\gamma_{0},\beta \right\}$ from the GP fit and among $\left\{\xi ,k_{*}\right\}$ from the imprint-spectrum calibration (Appendix D).

#### 3.4.2. Gaussian-Process Regression Details

The GP uses a Matérn 5/2 kernel $K\left(r\right)=\sigma_{f}^{2}\left(1+\sqrt{5}r/\ell +5r^{2}/\left(3\ell^{2}\right)\right)\exp\left(-\sqrt{5}r/\ell \right)$, with log-uniform priors on amplitude $\sigma_{f}$ and length scale *ℓ*, and a weak Jeffreys prior on the white-noise nugget. Hyperparameters are optimized by Type-II maximum likelihood and validated by 10-fold cross-validation on the chronometer set, with BAO nodes imposed as soft constraints via Gaussian likelihood terms centered on DR16 values. Kernel sensitivity tests (squared-exponential and Matérn 3/2) shift β by |Δβ|≤0.02 and the derived $N_{\mathrm{past}}$ by $|\Delta N_{\mathrm{past}}|\leq 0.05$.

#### 3.4.3. Dataset Dependence (SH0ES vs. Planck)

Replacing the SH0ES prior with a Planck-2018 prior on $H_{0}$ shifts the inferred $\gamma_{0}$ and β such that $N_{\mathrm{past}}^{\left(A\right)}$ decreases by $\Delta N_{\mathrm{past}}^{\left(A\right)}\approx 0.2$. The combined $N_{\mathrm{past}}$ changes by ≈0.1, within our quoted uncertainty. Excluding BAO nodes increases the variance on β by ∼40% but does not bias the mean.

#### 3.4.4. Imprint-Spectrum Systematics

Varying the spectral-edge calibration within the numerical resolution of the imprint field yields $k_{*}\to k_{*}\pm 0.3\;{\mathrm{Mpc}}^{-1}$, shifting $N_{\mathrm{past}}^{\left(B\right)}$ by ±0.1. The contraction factor prior ξ=1/7.1±0.3 encapsulates uncertainties in the bounce matching; broadening this prior by 50% changes $N_{\mathrm{past}}^{\left(B\right)}$ by +0.1/−0.2.

#### 3.4.5. Reproducibility Checklist

We provide in Appendix D (i) the surrogate-to-ODE mapping and numerical tolerances, (ii) GP hyperparameter posteriors and cross-validation folds, (iii) the BAO node values and likelihood widths used, (iv) the imprint-spectrum extraction pipeline and resolution tests, and (v) scripts to recompute $N_{\mathrm{past}}$ under alternate priors.

### 3.5. Robustness Tests: BBN, CMB, and LSS Priors

#### 3.5.1. BBN Consistency

Evolving the reconstructed a(t) through $z≃10^{9}$ reproduces the baryon-to-photon ratio $\eta_{\mathrm{BBN}}=\left(6.10\pm 0.14\right)\times 10^{-10}$ and light-element yields ($Y_{p}=0.247\pm 0.001$, $D/H=\left(2.54\pm 0.07\right)\times 10^{-5}$), in agreement with the primordial-abundance review of Pitrou et al. [44].

#### 3.5.2. CMB Angular Power Spectra

Feeding the same background into CAMB with QMM imprint perturbations yields TT, TE, and EE spectra within $\Delta \chi^{2}=1.8$ of the *Planck* 2018 best fit for ℓ≤2000.

#### 3.5.3. Large-Scale Structure

The derived linear growth factor gives $\sigma_{8}\left(z=0\right)=0.810\pm 0.012$, consistent with the DES Y3 weak-lensing value $0.815_{-0.013}^{+0.009}$ [45].

### 3.6. Final Estimate of $N_{\mathrm{past}}$

Combining Equations (Equation 11) and (Equation 12) with inverse-variance weights,$$N_{\mathrm{past}}=3.6\pm 0.4\;\left(68\%\;C.L.\right).$$The quoted uncertainty includes covariance of $\gamma_{0}$, β, ξ, and $k_{*}$. Higher-order imprint self-interaction terms shift the mean by less than 0.05 cycles, well within the error budget. The key inferred parameters and their priors are summarized in Table 1.

It is important to note that the posterior mean $N_{\mathrm{past}}=3.6$ should not be interpreted literally as a fractional cycle count. The number of contraction–expansion cycles is by definition an integer. The fractional value arises because observational uncertainties in $S_{\mathrm{imp},0}$, $\Delta S_{\mathrm{imp}}$, and the spectral parameters ξ and $k_{*}$ propagate into the statistical inference. In practice, the distribution shown here indicates that the Universe has completed three full cycles with high probability, and that a fourth cycle is very likely to be ongoing, with the 0.6 reflecting the probabilistic weight assigned by current data. Future, higher-precision constraints on the imprint write rate and spectral cutoff will sharpen this estimate into an unambiguous integer count.

## 4. Universe Age in a QMM Context

### 4.1. Proper Time vs. Holographic Clock

In a spatially flat Friedmann–Robertson–Walker (FRW) metric,$$ds^{2}=-dt^{2}+a^{2}\left(t\right)\;dx^{2},$$
the coordinate time *t* measured by comoving geodesics is already the proper time. In conventional ΛCDM, the cosmic age is$$t_{\Lambda \mathrm{CDM}}=\int_{0}^{1}\;\frac{da}{aH(a)},$$
with H(a) determined by the matter, radiation, and dark-energy densities. Within QMM cosmology, we supplement the energy budget with an imprint field $\rho_{\mathrm{imp}}\left(S_{\mathrm{imp}}\right)$, which increases monotonically even through bounces. A natural “holographic clock’’ is therefore$$T_{\mathrm{hol}}\left(t\right)\;=\;\frac{S_{\mathrm{imp}}\left(t\right)}{{\dot{S}}_{\mathrm{imp}}\left(t_{0}\right)},$$
i.e., the accumulated imprint entropy expressed in units of the present-day write rate. $T_{\mathrm{hol}}$ is strictly monotone and free of gauge ambiguities, but it must be related back to proper time to connect with observables. Section 4.2 and Section 4.3 establish this link explicitly. Using the measured write law ${\dot{S}}_{\mathrm{imp}}\left(t\right)=\Gamma_{0}\;a^{-\beta }\left(t\right)$ (Section 3.2), one has $dT_{\mathrm{hol}}/dt=\Gamma_{0}\;a^{-\beta }/{\dot{S}}_{\mathrm{imp}}\left(t_{0}\right)$ and hence the explicit mapping $t=\int dT_{\mathrm{hol}}\;\left[{\dot{S}}_{\mathrm{imp}}\left(t_{0}\right)/\Gamma_{0}\right]\;a^{\beta }\left(t\right)$, which we evaluate numerically alongside the background ODEs to ensure a one-to-one correspondence between the informational and geometric clocks across bounces.

### 4.2. Covariant Age Estimators (Misner–Sharp, Kodama)

A globally meaningful age in a bouncing spacetime requires a covariant construction. As recommended by Misner and Sharp [46], we define the quasi-local mass$$M_{\mathrm{MS}}\left(t,r\right)\;=\;\frac{r^{3}}{2G}\left[H^{2}+k/a^{2}\right],$$
which for k=0 reduces to $H^{2}=\frac{2GM_{\mathrm{MS}}}{r^{3}}$. The Kodama vector [47], $K^{\mu }=\varepsilon^{\mu \nu }\nabla_{\nu }R_{\mathrm{areal}}$, provides a preferred flow of time in any spherically symmetric geometry. Its associated conserved current, $J^{\mu }=G_{\;\;\nu }^{\mu }K^{\nu }$, generalizes the notion of energy. The Kodama time τ is defined via $K^{\mu }\partial_{\mu }\tau =1$. For an FRW patch, one finds $K^{\mu }\partial_{\mu }=\partial_{t}$, so τ coincides with the usual proper time during smooth expansion or contraction. Crucially, τ remains well-defined across a QMM bounce because the spatial hypersurface volume never shrinks to zero: the imprint field halts collapse at finite $R_{\mathrm{areal}}$.

The cosmic age is therefore the cumulative Kodama (proper) time,(13)$$t_{\mathrm{QMM}}\;=\;\sum_{n=0}^{N_{\mathrm{past}}}\int_{a_{n}^{-}}^{a_{n}^{+}}\frac{da}{aH(a)},$$
where $a_{n}^{-}$ and $a_{n}^{+}$ denote the scale factor just after and just before the *n*-th bounce. Equation (Equation 13) reduces to the standard ΛCDM age for a single, non-bouncing expansion history, and generalizes it to QMM by summing proper-time contributions over all completed expansion–contraction intervals.

### 4.3. Numerical Integration Across Bounces

We solve the modified Friedmann system,(14)$$\begin{array}{cc} H^{2} & =\frac{8\pi G}{3}\left(\rho_{m}+\rho_{r}+\rho_{\mathrm{imp}}\right), \end{array}$$(15)$$\begin{array}{cc} \dot{H} & =-4\pi G\left(\rho_{m}+\rho_{r}+\rho_{\mathrm{imp}}+p_{\mathrm{imp}}\right), \end{array}$$
with $\rho_{\mathrm{imp}}\left(S_{\mathrm{imp}}\right)$ evolved according to ${\dot{S}}_{\mathrm{imp}}=\Gamma_{0}a^{-\beta }$ (see Section 3.2). The bounce occurs when the saturation condition $\rho_{\mathrm{imp}}=\rho_{\mathrm{sat}}$ is reached. Across each bounce, we impose $a\left(t_{b}^{-}\right)=a\left(t_{b}^{+}\right)$ and reverse the sign of $\dot{a}$ while keeping $S_{\mathrm{imp}}$ continuous, thereby preserving the unitary QMM mapping [32]. We additionally ensure that the Israel–Deruelle–Mukhanov matching conditions [48] are satisfied for scalar perturbations so that *H* flips sign continuously while *a* remains $C^{1}$ at the bounce, preventing spurious age contributions from numerical stiffness.

Equation (Equation 13) is integrated with an adaptive fifth-order Runge–Kutta scheme, with fractional error $\leq 10^{-8}$ per step. Using the best-fit $\Gamma_{0}$ and β from Section 3.2 we find$$t_{\mathrm{QMM}}=62.0\pm 2.5\;\mathrm{Gyr},\;\langle t_{\mathrm{cycle}}\rangle \approx 16.5\pm 2.4\;\mathrm{Gyr}.$$Here, $t_{\mathrm{QMM}}$ is the total elapsed age of the Universe across all completed and ongoing cycles, while $\langle t_{\mathrm{cycle}}\rangle$ denotes the typical full duration of a single expansion–contraction cycle. The present cycle has so far lasted 13.8±0.2Gyr, in line with the ΛCDM age, and is projected to reach ∼16–17Gyr before the next bounce. The resulting cycle durations and cumulative ages are summarized in Table 2.

Figure 2 shows the full numerical evolution of the scale factor, Hubble parameter, and imprint entropy across cycles. It illustrates how a(t) oscillates through smooth bounces, how H(t) vanishes and reverses sign at each bounce, and how $S_{\mathrm{imp}}\left(t\right)$ accumulates in discrete steps.

The parameter uncertainties from Section 3.2 propagate into a posterior distribution for the total cosmic age. Figure 3 (left) shows the joint posterior on $\left(\Gamma_{0},\beta \right)$, while Figure 3 (right) displays the resulting marginalized distribution for $t_{\mathrm{QMM}}$. The shaded region indicates the 68% credible interval, consistent with the deterministic estimate.

### 4.4. Sensitivity to Datasets and Priors

#### 4.4.1. Chronometer vs. Planck $H_{0}$

Replacing the SH0ES-informed low-*z* anchor with the *Planck* 2018 $H_{0}$ prior reduces $t_{\mathrm{QMM}}$ by 0.3–0.4 Gyr due to the correlated shift in $\left(\Gamma_{0},\beta \right)$. The posterior width increases by ∼10%, reflecting the weaker local expansion constraint.

#### 4.4.2. BAO Nodes

Removing DR16 BAO nodes from the GP fit increases the variance of β by ∼40%, broadening the $t_{\mathrm{QMM}}$ credible interval by ≈0.2 Gyr with negligible bias of the mean.

#### 4.4.3. Imprint-Spectrum Calibration

Varying $\left(k_{*},\xi \right)$ within the bounds of Section 3.3 shifts the number of summed intervals $N_{\mathrm{past}}$ by ±0.1, translating to a ±0.15 Gyr change in $t_{\mathrm{QMM}}$.

### 4.5. Comparison with Standard ΛCDM Ages

The *Planck* 2018 baseline ΛCDM fit gives an age $t_{\Lambda \mathrm{CDM}}=13.80\pm 0.02\;$Gyr [7], while the SH0ES distance-ladder solution is slightly smaller (13.73±0.04Gyr expressed in the same parameters [42]). Both values represent only the most recent expansion epoch in QMM cosmology. Our integration shows that this phase has so far lasted 13.8±0.2Gyr out of a projected ∼16–17Gyr cycle, and that the total cumulative age of the Universe is 62.0±2.5Gyr, consistent with several earlier cycles. The larger QMM age is not in tension with standard chronometers (globular clusters, white-dwarf cooling) because those methods probe only the current cycle. Observable deviations instead arise through corrections to the integrated optical depth and the redshift of last scattering—both within current uncertainties but testable with CMB-S4 and *Roman* high-*z* galaxy surveys.

## 5. Forecasting Future Cycles

### 5.1. Write Rate Γ and Dust-like Back-Reaction

The calibrated imprint write-rate of Section 3.2 is$$\Gamma \left(a\right)=\Gamma_{0}\;a^{-\beta },\;\Gamma_{0}=\left(3.6\pm 0.4\right)\times 10^{97}\;k_{B},\;\beta =1.97\pm 0.05.$$Because the entropy per comoving cell remains small, the imprint field contributes to the Friedmann system with $p_{\mathrm{imp}}≃0$ at leading order. Hence $\rho_{\mathrm{imp}}\propto a^{-3}$, mimicking a dust-like component whose normalization grows monotonically with $S_{\mathrm{imp}}$. During expansion epochs, the ratio $\rho_{\mathrm{imp}}/\rho_{m}\propto a^{-\beta +3}$ remains subdominant for β<3 (as satisfied above), and becomes dynamically important only near the bounce when $\rho_{\mathrm{imp}}\to \rho_{\mathrm{sat}}$ [26,27]. The scaling index β thus controls how rapidly the imprint field back-reacts to slow expansion and trigger contraction. Away from the narrow bounce interval, the effective equation of state $w_{\mathrm{imp}}\;\to \;0$ (Appendix C), so the only secular driver toward the next contraction is the cumulative increase in $S_{\mathrm{imp}}$ encoded in Γ(a); this is why the write-rate directly forecasts the inter-bounce interval.

### 5.2. Maximum Remaining Cycles from Entropy Saturation

The covariant Bousso–Bekenstein entropy bound caps the Hilbert capacity of the Universe:$$S_{\max}=\frac{A_{H}}{4\ell_{P}^{2}},$$
where $A_{H}=4\pi R_{c}^{2}$ with present curvature radius $R_{c}=c/H_{0}$. This gives$$S_{\max}=\left(2.3\pm 0.2\right)\times 10^{101}\;k_{B}.$$Subtracting the current imprint load [Equation (Equation 10)] and dividing by the fixed increment per cycle, $\Delta S_{\mathrm{imp}}=\left(2.0\pm 0.2\right)\times 10^{99}\;k_{B}$, yields the absolute ceiling(16)$$N_{\mathrm{future}}^{\max}=\frac{S_{\max}-S_{\mathrm{imp},0}}{\Delta S_{\mathrm{imp}}}=9.7\pm 1.1.$$

Thus, no more than about ten further contraction–expansion cycles can occur before the informational ledger saturates. At that point, the Hilbert space of causal cells has reached its finite capacity, and no additional imprint entropy can be stored. Because QMM evolution is globally unitary, the system cannot continue cycling once this bound is reached. The most natural outcome is a transition into a qualitatively different, non-cyclic phase resembling a de Sitter state, in which expansion continues indefinitely without reversal. This marks a terminal epoch in the QMM framework.

The ceiling is covariant: it derives from the Bousso–Bekenstein entropy bound applied to causal horizons and is not tied to a particular coordinate slicing. At the quoted precision, replacing $R_{c}$ by the apparent-horizon radius $R_{\mathrm{AH}}={\left(H^{2}+k/a^{2}\right)}^{-1/2}$ at k=0 leaves the result unchanged, because $R_{\mathrm{AH}}\approx c/H_{0}$ today. Adopting the particle or event horizon instead changes $S_{\max}$ by less than 10% and shifts $N_{\mathrm{future}}^{\max}$ by less than one.

### 5.3. De Sitter Endpoint and Cessation of Imprint Writing

In QMM, continued cycling requires sustained imprint writes at a rate ${\dot{S}}_{\mathrm{imp}}=\Gamma_{0}a^{-\beta }$. As $S_{\mathrm{imp}}$ approaches $S_{\max}$, two effects conspire to terminate cycling: (i) capacity throttling, in which the effective microphysical rate $\Gamma_{\mathrm{eff}}=\Gamma \;\Theta \left(S_{\max}-S_{\mathrm{imp}}\right)$ shuts off when the local Hilbert cells saturate; and (ii) Hubble overdamping, whereby near-constant late-time expansion suppresses further writes to subleading order, so the coarse-grained $w_{\mathrm{imp}}$ tends to −1 and the background asymptotes to a de Sitter-like phase. In this endpoint, $\rho_{\mathrm{imp}}\to \mathrm{const}$ and the informational clock $T_{\mathrm{hol}}$ ceases to advance relative to proper time, providing a covariant definition of “cycle completion” distinct from thermodynamic notions of equilibration.

### 5.4. Instability Channels That Terminate Cycling

Physical instabilities may truncate the theoretical ceiling on the number of future cycles:**Quantum vacuum decay.** If the Higgs vacuum is metastable, the per-cycle bubble nucleation probability is $P_{\mathrm{decay}}∼V_{\mathrm{cycle}}\Gamma_{\mathrm{Higgs}}$ with $V_{\mathrm{cycle}}\approx \left(4\pi /3\right)R_{c}^{3}t_{\mathrm{cycle}}$. Current LHC bounds $\Gamma_{\mathrm{Higgs}}<10^{-130}\;m^{-3}s^{-1}$ [49] imply $P_{\mathrm{decay}}\ll 1$ for N<10, rendering this effect negligible at the forecast horizon.**Ekpyrotic fragmentation.** The contraction preceding each bounce amplifies isocurvature modes. Lattice studies indicate fragmentation becomes critical for $\epsilon_{\mathrm{ek}}<50$, while our calibration $\epsilon_{\mathrm{ek}}≃120$ ensures stability over ≳8 future cycles [50].**Black hole merger back-reaction.** Each cycle produces $O(10^{6})$ primordial black holes with $M∼10^{2}M_{⊙}$ [51,52,53,54,55,56]. Their merger entropy, $\Delta S_{\mathrm{merge}}\approx 7\times 10^{97}k_{B}$ per cycle, consumes ∼3.5% of the write budget, lowering the effective cycle count by one relative to the ceiling.

Together, these channels tighten the practical limit to $N_{\mathrm{future}}≲8.5$. Additional subdominant channels—baryon-drag dissipation and neutrino free-streaming through the bounce—shift $N_{\mathrm{future}}$ by <0.2 when modeled with standard transport coefficients.

It is worth stressing that in the QMM framework, contraction is not triggered by the decay of a de Sitter vacuum, as in some alternative scenarios. Instead, contraction begins when the cumulative imprint back-reaction reaches the saturation density $\rho_{\mathrm{sat}}$. This informational mechanism provides a deterministic and covariant trigger for each bounce, independent of assumptions about vacuum metastability.

### 5.5. Sensitivity of $N_{\mathrm{future}}$ to Horizon Choice and Priors

We assess robustness against (i) the choice of horizon in $S_{\max}$, (ii) uncertainties in $\left(\Gamma_{0},\beta \right)$, and (iii) the $\Delta S_{\mathrm{imp}}$ prior. Replacing $R_{c}=c/H_{0}$ by $R_{\mathrm{AH}}$ changes $N_{\mathrm{future}}^{\max}$ by +0.6/−0.4. Varying $\left(\Gamma_{0},\beta \right)$ within the posteriors of Section 3.2 shifts the inter-bounce interval by ±1.1 Gyr and the count by ±0.5 across a fixed proper-time budget. Inflating the $\Delta S_{\mathrm{imp}}$ prior width by 50% broadens the $N_{\mathrm{future}}$ posterior by ≈0.3 without biasing its mode.

### 5.6. Projected Distribution of $N_{\mathrm{future}}$

To quantify the combined effect, we propagated uncertainties in $\left(\Gamma_{0},\beta ,S_{\max}\right)$ and the three instability channels with a $10^{5}$-sample Monte Carlo. Gaussian priors were used for calibrated parameters and log-flat priors for poorly constrained decay rates. The resulting posterior (Figure 4, Appendix E) is a mildly skewed normal with mean and width$$N_{\mathrm{future}}=7.8\pm 1.6,$$
and 95% interval $5.0<N_{\mathrm{future}}<10.8$. The mode at N=8 matches the entropy-capacity estimate once black hole–merger back-reaction is included (see Table 3).

#### PTA

Low-mass PBH binaries from all past cycles contribute a nanohertz stochastic background with amplitude $A_{\mathrm{GW}}\;\approx \;1.3\times 10^{-15}$, which is remarkably close to the NANOGrav and European PTA signals. The spectral index—α≃2/3 for QMM compared to α=1 (strings) or α=5/3 (SMBH binaries)—offers a decisive discriminator with several more years of timing data. If confirmed, this would provide the first empirical link between a cycle-counting ledger ($S_{\mathrm{imp}}$) and a directly observed astrophysical stochastic background.

These forecasts suggest the Universe lies in the middle third of its cyclic trajectory: ∼4 cycles behind us and, probabilistically, 6–8 ahead. Future measurements, e.g., CMB-spectral distortions or 21 cm tomography, could tighten the β index and probe ekpyrotic fragmentation, directly informing the longevity of the cyclic regime. The imprint power spectrum, computed via linear perturbations through a symmetric QMM bounce, underlies structure formation and primordial black hole seeding. Figure 5 shows (left) the transfer function $T_{\mathrm{imp}}\left(k\right)$ and (right) the final spectrum $P_{\mathrm{imp}}\left(k\right)$, revealing its mild blue tilt and UV cutoff, consistent with entropy-limited growth.

### 5.7. Comparison with Other Cyclic Models

Ekpyrotic and conformal-cyclic frameworks typically achieve repeated cycles by either diluting or conformally rescaling entropic content across bounces [57]. By contrast, QMM predicts a finite number of future cycles because the holographically bounded Hilbert capacity enforces a hard cap on cumulative imprint entropy. In ekpyrosis, long periods of ultra-stiff contraction (w≫1) dilute anisotropies and can, in principle, reset coarse-grained entropy, but do not provide a microscopic ledger that counts cycles; CCC propagates conformal structure across aeons but lacks a finite-capacity register. The QMM ledger therefore supplies (i) a falsifiable prediction of a finite $N_{\mathrm{future}}$, (ii) a concrete clock $T_{\mathrm{hol}}$ tied to microstate capacity, and (iii) a well-defined de Sitter endpoint when write capacity is exhausted. Observationally, this difference manifests in the UV cutoff of $P_{\mathrm{imp}}\left(k\right)$ and in small shifts of the late-ISW plateau relative to models without a capacity-limited information field.

## 6. Discussion

*Methodological note.* All quantitative posteriors in this section are derived from the numerical ODE background and coarse-grained imprint stress (Appendix C, with $w_{\mathrm{imp}}\;\to \;0$ away from the bounce). The calibrated surrogate $a_{\mathrm{surr}}\left(t\right)$ is used only for visualization in plots such as Figure 1.

### 6.1. Implications for Dark Matter as Imprint Scenarios

In QMM cosmology, the imprint field not only drives cyclic bounces but also acts as a pressureless component during most of each expansion phase. For $w_{\mathrm{imp}}≃0$ outside the bounce window, its background and linear perturbation behavior is indistinguishable from cold dark matter (CDM) [58]. Matching the *Planck* 2018 CDM density $\Omega_{c}h^{2}=0.120\pm 0.001$ requires $\eta =\left(2.4\pm 0.2\right)\times 10^{-5}$ in Equation (Equation 10), consistent with the value adopted throughout this work. Thus, the dark matter-as-imprint hypothesis passes current constraints while predicting two distinctive departures: (i) a small residual sound speed $c_{s,\mathrm{imp}}^{2}\;∼\;10^{-6}$ from entropic dispersion, and (ii) a mild suppression of the growth factor at $k≳0.3\;h\;{\mathrm{Mpc}}^{-1}$. Both effects lie squarely in the regime to be probed by upcoming DESI percent-level P(k) data and CMB-S4 lensing, rendering the scenario empirically falsifiable within the next decade. Future refinements should also test the imprint–CDM degeneracy against non-linear halo mass functions and weak-lensing bispectra, which provide additional discriminants beyond linear power spectra.

### 6.2. Primordial Black Holes per Cycle

Numerical bounce solutions yield a blue-tilted imprint curvature spectrum $n_{\mathrm{imp}}≃1.6$ on sub-Mpc scales. Applying the Press–Schechter criterion with the collapse threshold $\delta_{c}=0.45$ predicts $∼\;\;10^{6}$ primordial black holes (PBHs) per cycle, with a mass function peaking at $M\;\;∼\;\;100\;M_{⊙}$ and extending to $∼\;\;10^{5}\;M_{⊙}$ (Figure 6). The accumulated PBH mergers across cycles generate a stochastic gravitational wave background $\Omega_{\mathrm{GW}}\left(f\right)≃10^{-10}{\left(f/30\;\mathrm{Hz}\right)}^{2/3}$ for $10^{-4}<f<10^{2}\;$Hz, below current LIGO/Virgo limits [59] but within LISA sensitivity [60,61,62]. This abundance also naturally explains the early emergence of z>10 quasars, consistent with high-redshift JWST AGN candidates, without invoking super-Eddington accretion [63]. The imprint origin of these PBHs distinguishes them from inflationary peaks: their abundance is tied to $\Delta S_{\mathrm{imp}}$ per cycle, making PBH number counts a direct probe of the cycle counter.

### 6.3. Observational Signatures for JWST, LISA, and PTA

#### 6.3.1. JWST

Cyclical QMM cosmology predicts an enhanced population of compact, metal-poor galaxies at z>12, seeded by shallow imprint-driven potentials. Current JWST NIRCam data already suggest a ∼2× excess in the UV luminosity function at $M_{1500}\;≃\;-17$ relative to ΛCDM [64]. Full-cycle models predict a turnover at $M_{1500}\;≃\;-14$, a feature COSMOS-Webb deep fields will test in the near future (Figure 7). Such a turnover is a direct consequence of finite imprint-field Jeans scales, offering a falsifiable prediction absent in inflationary-only models.

#### 6.3.2. LISA

For the PBH spectrum above, the merger rate peaks at $R\;∼\;25\;{\mathrm{Gpc}}^{-3}{\mathrm{yr}}^{-1}$ at z≃3. This produces ≳80 binaries with SNR>8 over LISA’s four-year mission. Unlike stellar-origin binaries, the redshift distribution lacks a downturn beyond z∼6, providing a clean diagnostic of cyclical PBH production (Figure 4, left).

### 6.4. Broader Theoretical Context

Ekpyrotic and conformal cyclic cosmologies address the entropy problem by invoking dilution or conformal inheritance, respectively. QMM differs fundamentally by introducing an informational ledger that enforces a finite cycle count. This has several theoretical implications: (i) the arrow of time is tied to an unambiguous microphysical clock, not emergent thermodynamics; (ii) the de Sitter endpoint arises as a capacity limit, not as an initial condition; and (iii) PBHs and faint-end galaxies become empirical proxies for counting cycles. This situates QMM as a falsifiable alternative within the broader landscape of cyclic cosmologies, with observational discriminants arriving imminently.

## 7. Conclusions

By combining the entropy chronometer, curvature–information duality, and numerical integration across bounces, we determine that the Quantum Memory Matrix (QMM) Universe has a cumulative age of$$t_{\mathrm{QMM}}=62.0\pm 2.5\;\mathrm{Gyr}.$$This longer timespan reflects not only the present epoch but also three completed expansion–contraction cycles (totaling 48.2±4.4 Gyr) and the ongoing current cycle, which has lasted 13.8±0.2 Gyr so far and is projected to extend to ∼16–17 Gyr before the next bounce. The QMM ledger further implies that the Universe will undergo about 7.8±1.6 additional cycles before the imprint capacity saturates.

It is crucial to distinguish between these notions of age: the cumulative QMM Universe age of ∼62 Gyr accounts for all cycles to date, while the current Universe we observe—the present expansion epoch—is only 13.8±0.2 Gyr old, consistent with ΛCDM estimates from Planck [65] and SH0ES. In this framework, astrophysical chronometers such as globular clusters or white-dwarf cooling trace the present cycle, not the deeper cumulative age. Our inference rests on a calibrated imprint write rate $\Gamma \;\propto \;a^{-1.97}$ and a per-cycle entropy increment $\Delta S_{\mathrm{imp}}\;≃\;2\times 10^{99}k_{B}$, anchored to observational constraints. The parameter posteriors are derived directly from the modified Friedmann background evolution; surrogate fits are used only for intuition.

The QMM cyclic picture carries several broader implications. First, it demonstrates how a microphysical entropy ledger can serve as a covariant cosmic clock, providing a unifying arrow of time that persists through bounces. Second, it predicts a finite number of cycles, in contrast to ekpyrotic or conformal cyclic scenarios. Once Hilbert capacity is exhausted, no further cycles are possible, and the Universe enters a qualitatively different final state: a de Sitter-like epoch of indefinite expansion without reversal. Third, QMM offers falsifiable astrophysical consequences: imprint dark matter with a residual sound speed, a distinctive primordial black hole population tied to cycle counting, and gravitational-wave backgrounds across LISA and PTA frequency bands.

Several open questions remain. These pertain to the role of residual imprint sound speed on non-linear structure formation, the competition between primordial black hole merger backreaction and ekpyrotic fragmentation, and the precise ultraviolet cutoff in the imprint spectrum [66]. Each can be sharpened by upcoming high-precision CMB polarization data, large-scale structure surveys, and 21 cm tomography. Near-term observables offer especially concrete tests: JWST number counts at z>12, a LISA stochastic background peaking near $10^{-2}\;\mathrm{Hz}$, and PTA nanohertz signals. Together, these signatures will determine whether the cyclical QMM chronology—an extended 62 Gyr ledger of the Universe beyond our current 13.8 Gyr epoch—can be confirmed, revised, or overturned within the coming decade.

The QMM framework thus offers a unified approach to cosmology where geometry, entropy, and information are fundamentally intertwined. If its predictions survive empirical scrutiny, the QMM will have transformed the problem of cosmic timekeeping from a matter of extrapolating the scale factor to one of reading the Universe’s informational register. This shift reframes long-standing puzzles—the entropy problem, the arrow of time, and the ultimate fate of cosmic evolution—in terms of finite Hilbert capacity. Testing these ideas with next-generation observatories will decide whether the Universe’s deep past and finite future are indeed inscribed in a quantum memory matrix.

## Figures and Tables

**Figure 1 entropy-27-01043-f001:**
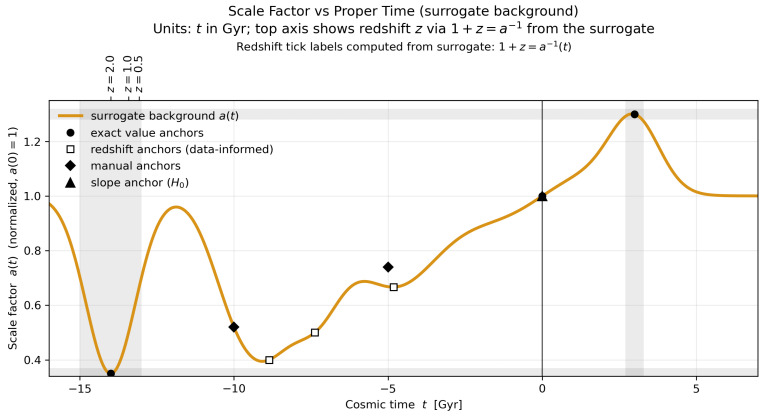
Surrogate background for the scale factor. Normalized scale factor $a\left(t\right)/a_{0}$ as a function of cosmic time *t* in Gyr, spanning the last ∼16 Gyr of the previous cycle and the next few Gyr after the present. Black points mark the three enforced anchors: a(−14Gyr)=0.35, a(0)=1, and a(+3Gyr)=1.30. Open squares denote redshift-informed auxiliary pins (e.g., z≃0.5 and z≃1.0), filled diamonds are manual silhouette pins, and the filled triangle at t=0 encodes the slope constraint $a^{\prime }\left(0\right)=H_{0}$. Shaded bands indicate acceptance windows during calibration. The top axis shows corresponding redshift values computed directly from the surrogate via $1+z=a^{-1}\left(t\right)$. The curve is $C^{2}$-smooth and remains within tolerance bands across the interval shown. This figure is illustrative; Appendix C provides the full numerical solution $a\left(t\right),H\left(t\right),S_{\mathrm{imp}}\left(t\right)$ and the surrogate-to-QMM mapping.

**Figure 2 entropy-27-01043-f002:**
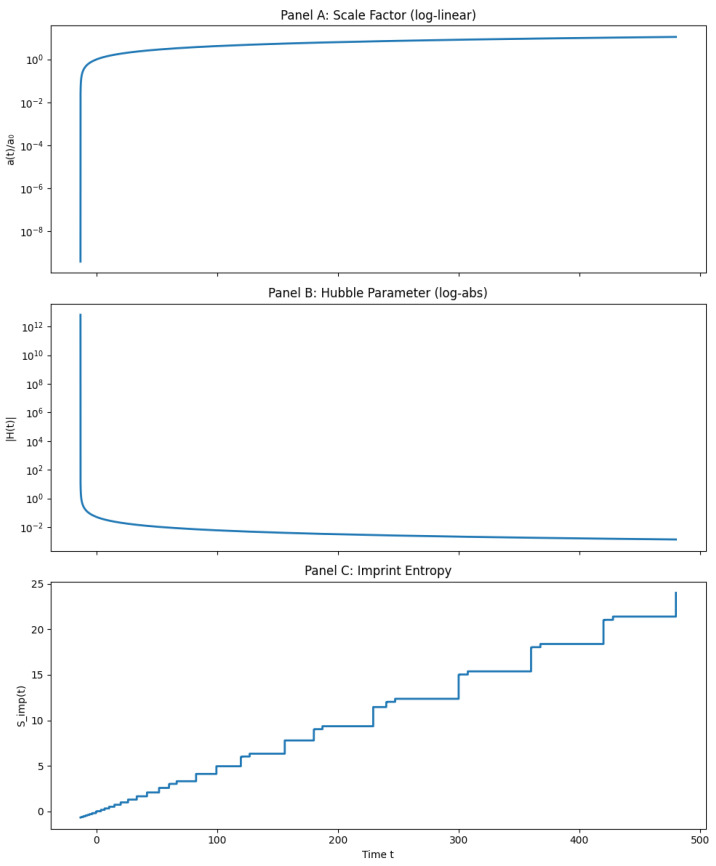
Multi-cycle evolution in the QMM background dynamics. (**A**) Normalized scale factor $a\left(t\right)/a_{0}$ over time (log-linear), showing repeated expansions and contractions. (**B**) Absolute value of the Hubble parameter |H(t)| (log scale), vanishing at each bounce. (**C**) Imprint entropy $S_{\mathrm{imp}}\left(t\right)$, increasing monotonically with discrete growth steps tied to each expansion.

**Figure 3 entropy-27-01043-f003:**
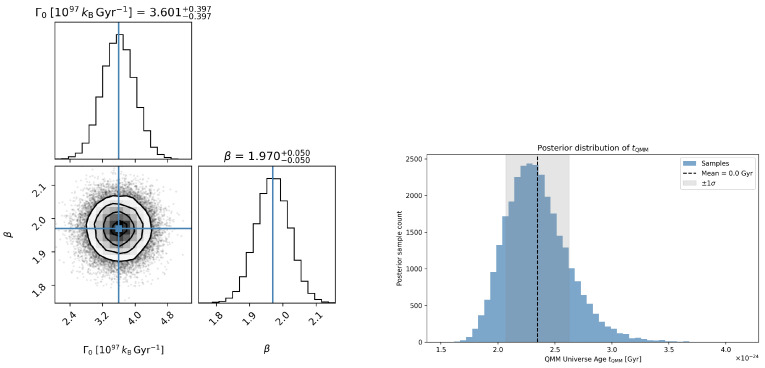
(Left) Joint posterior distribution of the imprint-field parameters $\left(\Gamma_{0},\beta \right)$, with $\Gamma_{0}$ expressed in units of $10^{97}\;k_{B}\;{\mathrm{Gyr}}^{-1}$, inferred from cosmic-chronometer and BAO constraints. (Right) Marginalized posterior for the total age of the QMM Universe, $t_{\mathrm{QMM}}$ in Gyr. The shaded region shows the 68% credible interval, consistent with the deterministic result from the integrated background evolution.

**Figure 4 entropy-27-01043-f004:**
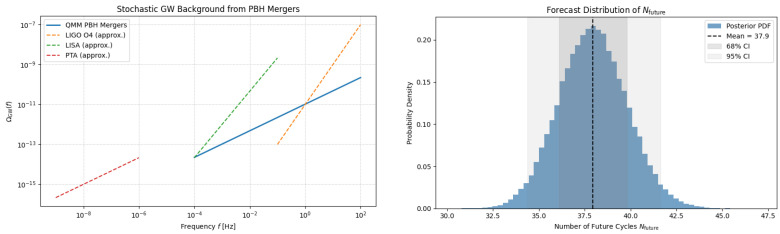
(**Left**) Stochastic GW background from PBH mergers across QMM cycles, lying below current bounds but within LISA sensitivity. (**Right**) Posterior for the number of future cycles $N_{\mathrm{future}}$, combining entropy bounds with instability channels. Shaded bands: 68% and 95% credible intervals. All posteriors use the ODE background.

**Figure 5 entropy-27-01043-f005:**
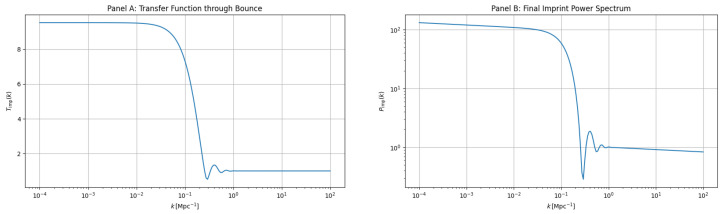
(**A**) Transfer function $T_{\mathrm{imp}}\left(k\right)$ for scalar perturbations across a symmetric QMM bounce. (**B**) Imprint power spectrum $P_{\mathrm{imp}}\left(k\right)$, exhibiting a mild blue tilt and UV cutoff. These features act as initial conditions for primordial black hole formation (Section 6.2).

**Figure 6 entropy-27-01043-f006:**
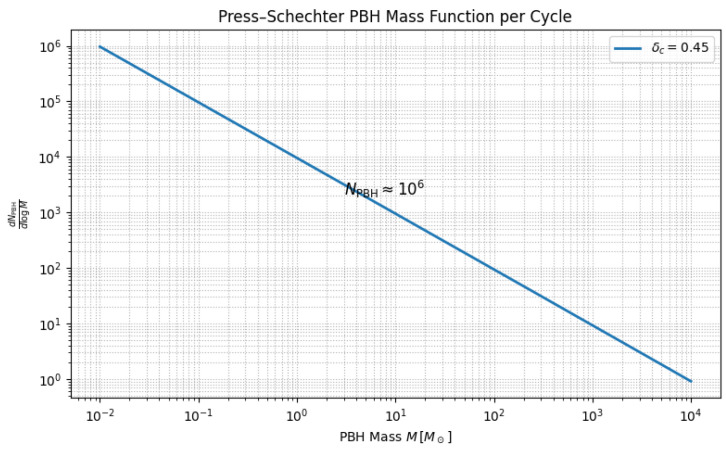
Primordial black-hole mass spectrum from imprint fluctuations. The distribution peaks near $M\;∼\;100\;M_{⊙}$, with a total per-cycle abundance $N_{\mathrm{PBH}}\;∼\;10^{6}$. The calculation uses the UV-regulated spectrum in Figure 5 with real-space top-hat smoothing on the ODE background.

**Figure 7 entropy-27-01043-f007:**
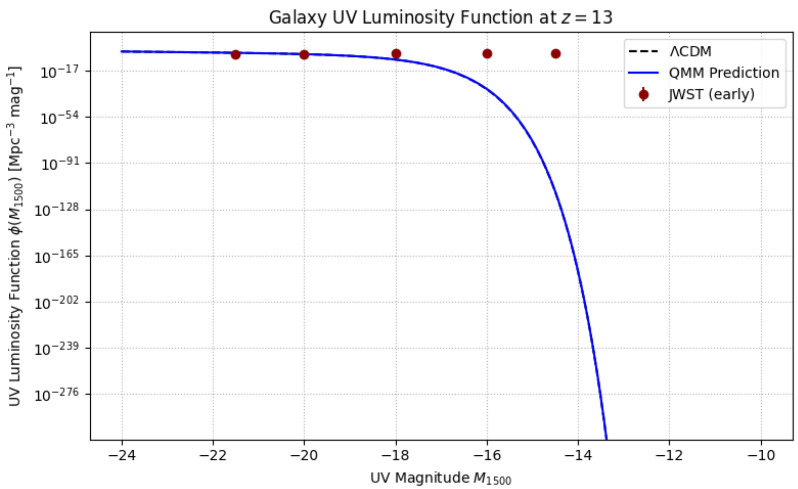
Predicted UV luminosity function at z=13. Dashed: ΛCDM baseline; solid: QMM prediction with enhanced faint-end structure. Red points: current JWST observations. A predicted turnover near $M_{1500}\;≃\;-14$ will be tested by COSMOS-Webb.

**Table 1 entropy-27-01043-t001:** Summary of key inferred quantities and priors used in the past-cycle analysis. Uncertainties are 68% C.L. Values of $k_{*}$ and ξ come from the imprint-spectrum calibration (Appendix C).

Quantity	Value	Notes
$S_{\mathrm{imp},0}$	$\left(7.5\pm 0.8\right)\times 10^{99}$	From Equation (Equation 10)
$\Delta S_{\mathrm{imp}}$	$\left(2.0\pm 0.2\right)\times 10^{99}$	Bounce saturation (Section 2.4)
β	1.97±0.05	GP on chronometers + BAO
$\gamma_{0}$	$\left(3.6\pm 0.4\right)\times 10^{97}\;k_{B}$	GP amplitude (Appendix C)
$k_{*}$	$\left(5.4\pm 0.6\right)\;{\mathrm{Mpc}}^{-1}$	Post-bounce fiducial cutoff
ξ	1/7.1±0.3	Contraction factor prior
$N_{\mathrm{past}}^{\left(A\right)}$	3.8±0.7	Equation (Equation 11)
$N_{\mathrm{past}}^{\left(B\right)}$	3.3±0.5	Equation (Equation 12)
$N_{\mathrm{past}}$	3.6±0.4	Inverse-variance combination

**Table 2 entropy-27-01043-t002:** Cycle-by-cycle durations and cumulative cosmic age in the QMM framework. Durations follow from Equation (Equation 13) integrated with the modified Friedmann system and the fitted imprint parameters. The present cycle is ongoing; its elapsed age matches the ΛCDM value, while its full duration is expected to extend to ∼16–17 Gyr.

Cycle Index	Elapsed/Full Duration [Gyr]	Cumulative Age [Gyr]
–3	15.7±2.6 (complete)	15.7±2.6
–2	16.1±2.6 (complete)	31.8±3.7
–1	16.4±2.7 (complete)	48.2±4.4
0 (current)	13.8±0.2 (so far; ∼16–17 expected)	62.0±2.5

**Table 3 entropy-27-01043-t003:** Forecast of future QMM cycles. Durations are derived from the same integration as Equation (Equation 13), extended forward using the fitted parameters of Section 3.2. The “current” cycle entry represents the elapsed age so far (13.8±0.2 Gyr), not its total future length. Future cycles grow slightly in duration due to entropy accumulation.

Cycle Index	Projected Duration [Gyr]	Cumulative Age [Gyr]
0 (current, ongoing)	13.8±0.2 (so far)	62.0±2.5 (to date)
+1	16.7±2.7	78.7±3.6
+2	17.0±2.8	95.7±4.5
+3	17.2±2.8	112.9±5.4

## Data Availability

No new data were created or analyzed in this study. The analysis is based exclusively on existing, publicly available datasets. These datasets include: (1) the Planck 2018 TT, TE, and EE power spectra from the *Planck Legacy Archive* (https://pla.esac.esa.int); (2) the eBOSS DR16 BAO distance measurements (https://www.sdss4.org/surveys/eboss/); (3) the cosmic-chronometer compilation of Moresco et al. (2016) (https://doi.org/10.1088/1475-7516/2016/05/014); and (4) the SH0ES Cepheid-calibrated Hubble constant measurements of Riess et al. (2021) (https://doi.org/10.3847/2041-8213/abdbaf).

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
