# Peer review of "Counting Cosmic Cycles: Past Big Crunches, Future Recurrence Limits, and the Age of the Quantum Memory Matrix Universe"

_entropy, 2025, doi:10.3390/e27101043_

Round 1

Reviewer 1 Report

Comments and Suggestions for Authors

In the manuscript the authors investigate the implications of the so-called Quantum Memory Matrix cosmology, and of a cyclical evolution of the Universe. The definition of a cycle is introduced in Eqs. (5)-(8), with the number of cycle given by Eq. (8). By using numerical methods, with the help of the Friedmann equations, the past and future number of cycles is obtained. Even that the content of the manuscript is highly speculative, it may be publishable in Entropy if the authors would fully consider the following points:

  1. It would be useful for the potential readers if for then sake of clarity  more information on the QNM and imprint entropy would be provided.
  2. At least intuitively one would expect for the number of cycles to be a positive integer, however, the numbers provided in this study are not integers. Is there any explanation for this result?
  3. What is the physical origin/nature of the imprint field?
  4. It would be useful to present the Figures in physical units. For example, what is the unit of time/entropy in Fig. 3?
  5. How the equation for \Gamma (a) at the beginning of Section 5.1 was obtained?
  6. What would happen with the Universe after all cyclic evolutions would end? What would be its final state?
  7. Is the second law of thermodynamics valid during all cycles? Does the entropy still increases during the contracting phase?
  8. The authors do not discuss the de Sitter and accelerated  phases of evolution. How would a de Sitter type expansion revert to a contraction? What would cause the physical process that initiates the transition from expansion to contraction, and vice-versa?
  9.  Is there any relation between the imprint entropy and the von Neumann entropy?
  10. The Introduction and the Conclusion Sections may be extended to add more discussions on the general context of the manuscript, and the significance of the obtained results.      

Reviewer 2 Report

Comments and Suggestions for Authors

See attached PDF.

Reviewer 3 Report

Comments and Suggestions for Authors

I think the paper is of good quality and in general well organized, some Figures can be shonw a bit better in order to see more clearly the details.

Reviewer 4 Report

Comments and Suggestions for Authors

The article proposes an interesting concept, expanding on past proposals of a cyclical universe that evolves through a series of bounces and contractions. An accumulating entropy, referred to as "imprint entropy", is argued to continually increase with each cycle, never decreasing during a collapsing phase. 3.6 +/- 0.4 past cycles are predicted as well as 7.8 +/- additional cycles beyond the current. Arguments justifying these predictions are provided, but these do not seem that convincing. Further explanation and clarifications would be helpful to the reader. Likewise for justification of an absolute limit to these "imprint entropy" increasing cycles. I am not convinced of that limit either. A proposed ultimate non-cycling ongoing de Sitter stage following the last allowed imprint entropy stage would not halt general entropy increase. I don't see why it would halt imprint entropy either. The imprint entropy is simply defined in eq. (1), but is not justified. Physical justification and explanation are warranted. Geometry-INformation Duality (GID) is stated to posit a one-to-one map between imprint entropy and the effective curvature radius of the spatial hypersurface, with the curvature radius inversely proportional to an imprint density proportional to imprint entropy per volume. Adequate discussion of this duality is lacking. Figure 1, providing a multi cycle evolution of the scale factor through past cycles, and the next is interesting, but the underlying physics generating this is also missing. Overall, the proposal is interesting, but insufficiently explained. The text should be significantly expanded before warranting publication.

Round 2

Reviewer 1 Report

Comments and Suggestions for Authors

The authors have improved their manuscript, and hence I think the present version is suitable for publication in Entropy.

Reviewer 4 Report

Comments and Suggestions for Authors

In the current revision, the authors have resolved all of my prior concerns. I now support its publication.